# Rice as a Determinant of Vietnamese Economic Sustainability

**Kamil Maitah [1], Luboš Smutka [1] , Jeta Sahatqija [2], Mansoor Maitah [2],\***
**and Nguyen Phuong Anh [2]**

[1] Department of Trade and Finance, Faculty of Economics and Management, Czech University of Life Sciences in Prague, Kamycka 129, 16500 Prague, Czech Republic; maitahk@pef.czu.cz (K.M.); smutka@pef.czu.cz (L.S.)

[2] Department of Economics, Faculty of Economics and Management, Czech University of Life Sciences in Prague, Kamycka 129, 16500 Prague, Czech Republic; sahatqija@pef.czu.cz (J.S.); 1nguyen@centrum.cz (N.P.A.)

**\*** Correspondence: maitah@pef.czu.cz

**Abstract:** This paper aims to examine the rice industry in Vietnam during the period 1997–2017, focusing its production and export. The total area of Vietnam is 33.121 million hectares, out of which 39.25% consists of agricultural land. The agricultural sector adds up to 24% of the gross domestic product (GDP), 20% of the total exports and over 70% of the total employment. Vietnam's economy is highly dependent on the agricultural sector, specifically rice production, which constitutes 30% of the country's total agricultural production value. While its production at first aimed to ensure food security in the country, to date, Vietnam is one the world's largest exporters. While extensive research has explored the rice industry, studies looking at the production through the use of fertilizers, external factors such as the exporting price of other countries and world consumption rates are still lacking. Given the complexity of the topic, data were analyzed through descriptive, econometric and quantitative methods. For production and export analyses, two and four hypotheses were derived and examined, respectively, all based on economic theory. The model consisted of two equations: (i) the paddy production is impacted by rice's yield and fertilizer use and (ii) in addition to internal factors, the growth of exporting rice in Vietnam depends also on external factors such as Thailand's rice export price and world consumption rates. Based on the model, a dynamic forecasting method was employed, using the previous forecast values of the dependent variables to compute the future ones. Findings showed that 98% of Vietnam's rice production is explained through the yield and fertilizer usage and 83% of Vietnam's rice export is explained by the production, the price in Vietnam and Thailand and the consumption levels around the world. When it comes to forecasting, an 8% growth is predicted with a peak in quantity produced, with 49,461 thousand tons in 2023, yet with difficulties when it comes to exporting. The research predicts a stagnation in exports.

**Keywords:** rice; Vietnam; agriculture; foreign trade

## 1. Introduction

Vietnam is one of the world's richest agricultural regions, with a very strong agricultural groundwork. The agricultural sector plays an important role in the country's economy, contributing 24% of the GDP and generating 20% of export revenues. Over 70% of the national labor force is employed within this sector [1,2]. Vietnam's agricultural growth, and that of the rice sector in particular, has set the foundations for the country's overall economic success. In Vietnam, rice is also referred to as "white gold". Rice production is critical to Vietnam's agricultural industry, constituting 30%

of the country's total agricultural production value [3]. Nearly half of the total agricultural land is designed for rice production. Most rice grown in Vietnam is wet rice, typically grown in flooded fields and thus requiring a constant supply of water. Depending on the seasonal climate conditions, wet rice is harvested two or three times a year, offering maximum yields. Vietnam's growth track is characterized by the production of low quality rice. Low-quality rice was an immediate solution for resolving widespread hunger in the country. It soon became a strategic crop for national food security. Over time, improvements in the productivity of the country significantly contributed to the reduction of poverty. In addition to being a symbol of Vietnamese culture, rice is also a staple food for the majority of Vietnam's population. It is also used to produce rice wine, rice noodles, rice vinegar and rice crackers. With large quantities in stock, selling at a low price and with low production costs, Vietnam soon became one of the top five rice exporting economies in the world [4].

Vietnam marked a major breakthrough in rice production in 1989, with a record output of 19 million tons. In the 1990s, the growth rate of rice production was at an average of 5.6% per year, driven by increases in yield and planted areas [5]. Due to continuous expansion and production growth over the past decade, Vietnam has become one of the world's largest rice exporters; of course, this is after ensuring adequate supply for domestic consumption. Most of the rice produced in Vietnam was exported to the Asian, African and Middle Eastern markets [6].

The total harvested area of paddy fields is over 7.7 million hectares, with the largest proportions of harvested areas being in the Mekong River Delta and Red River Delta regions. The Mekong Delta alone is home to 20% of Vietnam's population, and it is responsible for over 50% of Vietnam's rice production and more than 90% of its rice export [7]. Moreover, 80% of Mekong Delta's population is engaged in rice cultivation and 78% of the delta's land is used for rice production, with more than 1600 rice varieties cultivated. Being known as the "rice bowl" of Vietnam, the annual rice production of the Mekong River Delta is almost three times that of the Red River Delta. The application of mechanization in rice production has helped the Mekong Delta to increase the region's productivity to 5.7 tons in 2013 per hectare from 4.3 tons per hectare in 2001, and the output increased to 24.5 tons from 16 tons initially. The higher productivity of the Mekong Delta underlines its importance in rice production for surpluses and food security within Vietnam. The possibility of surpluses in other areas of Vietnam remains highly dependent on stable climatic conditions, as farmers often face a food deficit for many months per year [8]. As seen in the Figure 1, about 54% of Vietnam's rice is cultivated in the Mekong River Delta and another 17% in the Red River Delta.

While rice production is highly dependent on the climatic conditions, one should bear in mind that the Vietnamese rice sector is also facing and dealing with severe environmental issues; therefore, currently, the industry is in search of more sustainable production methods [8]. Over the past decade, strategies for the increase of production have mainly included the use of pesticides. While it is true that the use of fertilizers has led to a visible increase in production and the ability to meet international demand, their overuse has led to pest and disease outbreaks [9]. Rice production has, for decades, served as an economic catalyst, thus enabling growth in rural areas and lowering poverty. Today, the rising costs of fertilizers and labor are having a visible influence on the overall production [8]. Although extensive research has been conducted in the rice industry, studies looking at the issues of production through fertilizer use, external factors, such as the exporting prices of other countries, and world consumption rates are still lacking. This paper aims to examine the rice industry in Vietnam during the period 1997–2017, placing special attention on its production and export. To further shed light on the matter, it forecasts the industry's growth. The country is primarily chosen because of the lack of sustainable rice production practices and the immediate need for the introduction of radical environmentally friendly technologies and labor know-how.

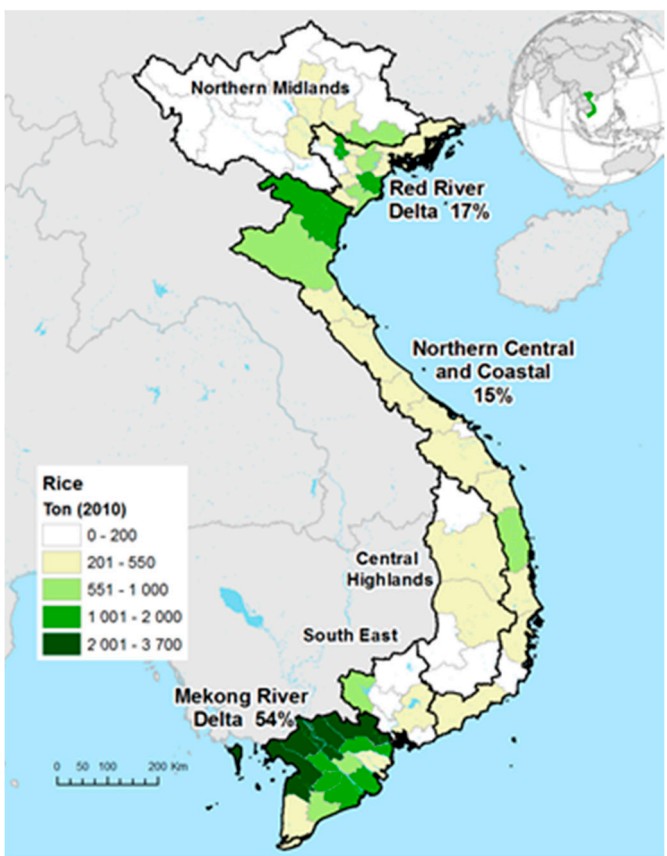

**Figure 1.** Rice cultivation in Vietnam. Source: USDA Foreign Agricultural Service.

## 2. Objectives and Methodology

The aim of this article is to analyze the rice industry in Vietnam during the period 1997–2017, paying special attention to production and export. For this purpose, descriptive, econometric and quantitative methods of data analysis are utilized. Specifically, for the analysis of production, we examine two hypotheses based on economic theory

**Hypothesis 1 (H1).** *Vietnam's rice production is impacted by paddy yield in a positive way.*

**Hypothesis 2 (H2).** *Vietnam's rice production is affected by the use of fertilizers, with a positive direction of change.*

In the case of export analysis, four hypotheses based on economic theories are examined:

**Hypothesis 3 (H3).** *The rice production in Vietnam has an impact on the export volume of Vietnam's rice; particularly, its changes are positive.*

**Hypothesis 4 (H4).** *The export price of Vietnam's rice affects the amount of Vietnam's rice export with a negative interaction.*

**Hypothesis 5 (H5).** *The exporting price of Thailand's rice has an impact on the quantity of Vietnam's rice export in a positive direction.*

**Hypothesis 6 (H6).** *The worldwide consumption of rice influences the export quantity of Vietnam's rice with a positive direction of change.*

The model consists of two equations. Firstly, the paddy production is impacted by rice's yield and fertilizer use. Secondly, the growth in the export of rice in Vietnam depends not only on internal factors, such as its export price and the status of paddy production, but also on international factors, such as the price of Thailand's rice export and the consumption of rice around the world. The economic model can be written as shown below:

$$y1 = \text{fce}\ (x2, x3)$$
$$y2 = \text{fce}\ (y1, x4, x5, x6) \tag{1}$$

where:

y1—The amount of rice production in Vietnam.
y2—The amount of exporting 5% broken rice in Vietnam.
x2—The yield of paddy in Vietnam.
x3—Fertilizer consumption in the production of rice in Vietnam.
x4—Exported price of 5% broken rice in Vietnam.
x5—Exported price of 5% broken rice in Thailand.
x6—Consumption of rice around the world.
fce—Function of

Considering the assumptions resulting from economic theory and the given economic model, the simultaneous equations model was chosen to determine the two dependent variables and their corresponding independent variables [10,11]. In contrast to the current research conducted in the field, this study does not include cultivated land among the variables in the model. The decision not to include this was impacted by the fact that the land in Vietnam continues to be under the sole ownership of the state. Despite the fact that, since 2012, farmers may lease cultivated land for a period of up to 50 years, the possibility that the government may take it back is always existent. In practice, this leads to the lack of motivation on the farmer's end to introduce any significant improvements to the land. In addition, the Land Law of 2003, specifically Article 70, stipulates a maximum assignment quota of 3 ha per annual crop, therefore discouraging farmers from investing in irrigation or new land cultivation methods. Such limitations have forced farmers to migrate to urban areas in search of other means of livelihood. The foundation of this model is to set up the production quantity of Vietnam's rice as the first dependent variable that is determined by the independent variables in the first equation. At the same time, the production quantity of Vietnam's rice plays the role of an independent variable in the second equation, among the other independent variables. The second independent variable is the exported quantity of Vietnam's rice. This can be mathematically written as follows:

$$\beta 11 y1 = \gamma 11 x1 + \gamma 12 x2 + u1$$
$$\beta 12 y2 = \beta 11 y1 + \gamma 13 x3 + \gamma 14 x4 + \gamma 15 x5 + \gamma 16 x6 + u2 \tag{2}$$

where was added:

β11, β12—parameters of dependent variables.
γ11 . . . γ16—parameters of independent variables.
u11, u12—random variables.

On one hand, the dependent variables are those whose values are generated and explained within the model. On the other hand, the independent variables are determined outside the model and explain the dependent variables. The random variable plays an important role in the model because it includes all other influences that affect the dependent variables and, for various reasons, these influences are not included in the model [12]. The variables' parameters are estimated using the least squares method. It is used to determine the line of best fit for the data set by minimizing the sum of the squares of errors between the data and the model. Just like variables from the data set, random variables are described

by measures of central tendency and measures of variability. Along with the quantification of the model, economic, statistical and econometric verifications are performed. In conclusion, based on the model, the forecasts of dependent variables are derived for the next six years (2018–2023). For this purpose, a dynamic forecasting method has been employed, where the additional lags of the dependent variables are added as regressors, and the previously forecasted values of the lagged variables are used in forming a forecast [13]. In other words, the dynamic forecast uses the value of the previous forecasted values of the dependent variables to compute the next ones.

## 3. Production and Export

In 2017, Vietnam was the world's fifth largest rice producing country [14]. The rice production has continuously increased from 27 million tons in 1997 to almost 46 million tons in 2017, an increase of 66% within a period of twenty years. This increase is attributed to the expansion of the rice area harvested, a higher yield and the increased use of pesticides and fertilizers. The rice area harvested expanded from 7.1 million hectares in 1997 to 7.7 million hectares in 2017; however, the annual growth was only 0.5% from 2005 to 2010 [7]. The harvested area of rice peaked in 2013 at 7.9 million hectares. Rice yields have nearly quadrupled since the 1970s [7]. Nowadays, yields of rice in Vietnam are the highest in the region and among the highest in the world. However, this comes at a cost. At 297 kg per hectare, Vietnam has the highest density of fertilizer use among the countries in the region. Other Southeast Asian countries apply an average of 156 kg per hectare of rice area [8]. The yields have remained high, holding a 10-year average (2006–2016) of 5.4 tons per hectare. Vietnam's yield greatly surpasses that of Thailand, another large rice producer in the region, with a 10-year average yield of only 3 tons per hectare.

In 2017, Vietnam was the third largest rice exporter in the world, behind India and Thailand, with an annual rice export volume which accounted for 15% of the world's total figure [6]. Furthermore, the rice export revenue amounted to 1.6 billion US$ and accounted for 12% of the GDP [2]. As shown in Figure 2, the exported amount had increased from 3.8 million tons in 1997 to 7.7 million tons in 2011, an increase of 104% within the fourteen-year period. However, since then, the exports dropped by 12% to 6.8 million tons in 2017. While discussing the volume of production, labor productivity is a key variable at the core of many studies conducted in the field and in the current literature. However, in the case of Vietnam, the agricultural value added per worker is growing at one of the slowest rates in the region. The low growing rate could be explained by the use of inefficient production methods but also the large amount of low-skilled labor. In theory, as well as in practice, the low education level is a constant obstacle in improving the agricultural productivity and movement up the supply chain.

Vietnamese rice has been exported to more than 150 nations around the world, with its market having been expanded to African and Middle Eastern countries. Its main export markets include China, Indonesia and the Philippines. China alone buys one-third of Vietnam's rice exports. Japan is another major rice importer in the region, but most of the grains produced in Vietnam fail to meet its strict quality standards due to the excessive use of fertilizers and pesticides; therefore, the country regularly imports 50% of its rice demand from the US. This reflects the fact that Vietnam still relies on traditional markets and has not focused on high-quality rice development. In general, the Vietnamese high-quality rice is exported to rather difficult markets, such as Japan and the EU, while lower-quality rice is exported to Asian and African markets and some countries in Africa [6].

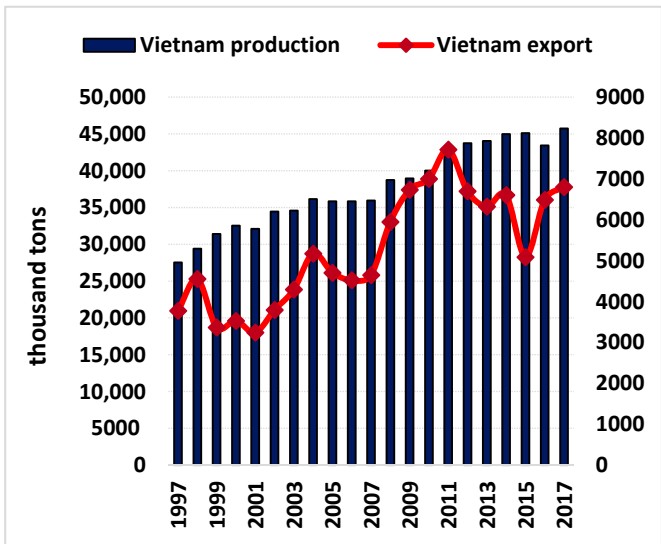

**Figure 2.** Vietnam's rice production and export for the period of 1997–2017. Source: Own processing, International Rice Research Institute.

## 4. Statistical–Mathematical Model

Using a statistical–mathematical model, the total Vietnamese rice production was examined in relation to its yield and the use of fertilizer. In addition, the examination included an analysis of how the export quantity of Vietnamese rice was impacted by specific internal and external factors. Among the many well-known types of rice exported from Vietnam, 5% broken rice was selected to be the subject of this study based on its popularity in the market. The data including units are shown in Table 1.

**Table 1.** Original dataset.

| Year | Vietnam Production | Vietnam Yield | Fertilizer Utilization | Vietnam Export | Vietnam Price | Thailand Price | World Consumption |
|---|---|---|---|---|---|---|---|
| | Thousand Tons | ton/ha | kg/ha | Thousand Tons | USD/ton | USD/ton | Thousand Tons |
| | $y_1$ | $x_2$ | $x_3$ | $y_2$ | $x_4$ | $x_5$ | $x_6$ |
| 1997 | 27,524 | 3.9 | 260 | 3776 | 256 | 304 | 375,312 |
| 1998 | 29,408 | 4.0 | 322 | 4555 | 288 | 304 | 382,346 |
| 1999 | 31,394 | 4.1 | 343 | 3370 | 228 | 248 | 394,210 |
| 2000 | 32,530 | 4.2 | 366 | 3528 | 183 | 202 | 396,030 |
| 2001 | 32,108 | 4.3 | 287 | 3245 | 148 | 173 | 399,045 |
| 2002 | 34,447 | 4.6 | 305 | 3795 | 187 | 192 | 403,173 |
| 2003 | 34,569 | 4.6 | 342 | 4295 | 167 | 198 | 403,695 |
| 2004 | 36,149 | 4.9 | 404 | 5174 | 224 | 238 | 410,062 |
| 2005 | 35,833 | 4.9 | 292 | 4705 | 239 | 286 | 415,823 |
| 2006 | 35,850 | 4.9 | 300 | 4522 | 249 | 305 | 423,043 |
| 2007 | 35,943 | 5.0 | 353 | 4649 | 294 | 326 | 430,931 |
| 2008 | 38,730 | 5.2 | 306 | 5950 | 614 | 650 | 448,300 |
| 2009 | 38,950 | 5.2 | 408 | 6734 | 432 | 555 | 453,985 |
| 2010 | 40,006 | 5.3 | 323 | 7000 | 416 | 489 | 459,396 |
| 2011 | 42,398 | 5.5 | 310 | 7717 | 505 | 543 | 466,672 |
| 2012 | 43,738 | 5.6 | 352 | 6700 | 397 | 563 | 471,356 |
| 2013 | 44,039 | 5.6 | 490 | 6325 | 363 | 506 | 475,659 |
| 2014 | 44,974 | 5.8 | 435 | 6606 | 377 | 423 | 471,780 |
| 2015 | 45,105 | 5.8 | 439 | 5088 | 334 | 386 | 466,512 |
| 2016 | 43,437 | 5.6 | 439 | 6488 | 347 | 396 | 476,740 |
| 2017 | 45,728 | 5.9 | 441 | 6800 | 472 | 398 | 480,485 |

Source: Own processing, International Rice Research Institute.

The main goal of the regression analysis is to isolate the relationship between each independent variable and the dependent variables. Multicollinearity is a state of very high intercorrelations among the independent variables [15,16]. It is therefore a type of disturbance in the data, and if present in the data, the statistical inferences made about the data may not be reliable [16]. It is necessary to check the multicollinearity in pairs of independent variables in each equation, meaning that the assumed independent variable could stay if its correlation coefficient with other independent variables is not higher than the accepted level of 0.85. The outcome of all pairs is demonstrated by the matrix of correlation in Table 2.

**Table 2.** Correlation matrix.

| y1 | x2 | x3 | x4 | x5 | x6 | |
|--------|--------|--------|--------|--------|--------|-----|
| 1.0000 | 0.9873 | 0.6979 | 0.6401 | 0.6481 | 0.9752 | y1 |
| | 1.0000 | 0.6351 | 0.6803 | 0.6838 | 0.9759 | x2 |
| | | 1.0000 | 0.2433 | 0.2648 | 0.6455 | x3 |
| | | | 1.0000 | 0.9363 | 0.7348 | x4 |
| | | | | 1.0000 | 0.7560 | x5 |
| | | | | | 1.0000 | x6 |

Source: Own processing, Gretl.

Table 2 shows four correlation coefficients higher than 0.85. As the result, the multicollinearity happens to the pair of X2 and Y1, X6 and Y1, X6 and X2, X5 and X4. To eliminate multicollinearity, X4 and X6 will be replaced by the first difference method, which are d_ X4 and d_ X6, respectively. The results are presented in Table 3.

**Table 3.** First differenced variables.

| Year | Vietnam Price | World Consumption |
|------|--------------|-------------------|
| | USD/ton | Thousand Ton |
| | d_x4 | d_x6 |
| **1998** | 32 | 7034 |
| **1999** | −60 | 11,864 |
| **2000** | −45 | 1820 |
| **2001** | −35 | 3015 |
| **2002** | 39 | 4128 |
| **2003** | −20 | 522 |
| **2004** | 57 | 6367 |
| **2005** | 15 | 5761 |
| **2006** | 10 | 7220 |
| **2007** | 45 | 7888 |
| **2008** | 320 | 17,369 |
| **2009** | −182 | 5685 |
| **2010** | −16 | 5411 |
| **2011** | 89 | 7276 |
| **2012** | −108 | 4684 |
| **2013** | −34 | 4303 |
| **2014** | 14 | −3879 |
| **2015** | −43 | −5268 |
| **2016** | 13 | 10,228 |
| **2017** | 125 | 3745 |

Source: Own processing, Gretl.

## 5. Verification of the Model

The task is to verify the estimated model in terms of economic theory, statistical significance and econometric conditions. This reveals how the applied economic theory fits the estimated model and it

furthermore shows how the results are statistically significant. Firstly, the value of the parameters is estimated by the two-stage least square method. Tables 4 and 5 show the results from each equation of the model.

**Table 4.** Parameters' estimation in the first equation.

| Variable | Coefficient | Std. Error | Std. Error | p-Value |
|---|---|---|---|---|
| Const | $-5.55187 \times 10^6$ | $1.53647 \times 10^6$ | $-3.613$ | 0.0003 |
| x2 | $7.94838 \times 10^6$ | 365,526 | 21.75 | $7.70 \times 10^{-105}$ |
| x3 | 10,107.7 | 3547.01 | 2.850 | 0.0044 |
| Mean Dependent var | 38,266,747 | | S.D. dependent var | 5,153,701 |
| Sum Squared resid | $1.02 \times 10^{13}$ | | S.E. of regression | 773,855.3 |
| R-squared | 0.979827 | | Adjusted R-squared | 0.977453 |

Source: Own processing, Gretl.

**Table 5.** Parameters' estimation in the second equation.

| Variable | Coefficient | Std. Error | Std. Error | p-Value |
|---|---|---|---|---|
| const | $-1.78648 \times 10^6$ | $1.7034 \times 10^6$ | $-1.048$ | 0.2946 |
| y1 | 0.139953 | 0.0536232 | 2.610 | 0.0091 |
| x4 | $-511.656$ | 1751.08 | $-0.2922$ | 0.7701 |
| x5 | 4603.82 | 1852.33 | 2.485 | 0.0129 |
| x6 | 0.0189398 | 0.0465265 | 0.4071 | 0.6840 |
| Mean Dependent var | 5,362,300 | | S.D. Dependent var | 1,374,057 |
| Sum Squared resid | $6.14 \times 10^{12}$ | | S.E. of Regression | 639862.1 |
| R-squared | 0.828856 | | Adjusted R-squared | 0.783217 |

Source: Own processing, Gretl.

When it comes to the interpretation of a regression coefficient, this represents the mean change in the dependent variable for each unit of change in an independent variable when all of the other independent variables remain unchanged [16].

In economic verification, the obtained estimates are examined if they are in accordance with the expectations of the theory that is being tested. The first equation representing Vietnam's rice production (y1), depending on Vietnam's rice yield (x2) and Vietnam's fertilizer use (x3) in the production is as follows:

$$y1 = -5,551,870 + 7,948,380x2 + 10,107.7x3 \tag{3}$$

- If yield of paddy in Vietnam increases by 1 ton per hectare, the amount of rice production increases by 7,948,380 tons.
- If fertilizer use in the rice production in Vietnam increases by 1 kg per hectare, the amount of rice production increases by 10,108 tons.

The second equation representing Vietnam's rice export (y2), depending on Vietnam's rice production (y1), Vietnam's rice price (x4), Thailand's rice price (x5) and the world's rice consumption (x6) is as follows:

$$y2 = 0.139953y1 - 1,786,480 - 511.656d\_x4 + 4603.82x5 + 0.0189398d\_x6 \tag{4}$$

- If the amount of rice production in Vietnam increases by 1000 tons, the exported amount of rice increases by 140 tons.
- If the exported rice price in Vietnam increases by 1 USD per ton, the exported amount of rice decreases by 512 tons.

- If the exported rice price in Thailand increases by 1 USD per ton, the exported amount of Vietnam's rice increases by 4604 tons.
- If rice consumption in the world increases by 1000 tons, the exported amount of Vietnam's rice increases by 19 tons.

The findings of the first and second equations are consistent with the theory's hypotheses. The statistical verification examining the significance of the model and the significance of the coefficients is next.

The coefficient of determination (denoted by R2) is a goodness-of-fit measure which represents the proportion of the variance in the dependent variable that is predictable from the independent variables [10]. In general, the higher the R2, the better the model fits the data. In the case of the first equation, the R2 is 0.979827, which means that 98% of Vietnam's rice production is explained by the yield and fertilizer use. The R2 of the second equation is 0.828856, which means that 83% of Vietnam's rice export is explained by the production, the price in Vietnam and Thailand and the consumption of rice around the world.

While the coefficient of determination provides an estimate of the strength of the relationship between the model and the independent variables, it does not provide a formal hypothesis test for this relationship. The *p*-value determines whether this relationship is statistically significant [12]. The null hypothesis (denoted by H0) states that there is no relationship between the variables; in other words, one variable does not affect the other. The level of statistical significance (denoted by $\alpha$) is expressed as a *p*-value between 0 and 1. The smaller the *p*-value, the stronger the evidence to reject the null hypothesis. A *p*-value less than or equal to the significance level $\alpha$ 0.05 is statistically significant [13]. The results are shown in the Tables 6 and 7 respectively.

**Table 6.** Parameters' significance in the first equation.

| Parameter | *p*-Value | Significance Level | Hypothesis |
|---|---|---|---|
| $\gamma12$ | $7.70 \times 10^{-105}$ | 0.05 | Reject H0 |
| $\gamma13$ | 0.0044 | 0.05 | Reject H0 |

Source: Own processing, Gretl.

**Table 7.** Parameters' significance in the second equation.

| Parameter | *p*-Value | Significance Level | Hypothesis |
|---|---|---|---|
| $\beta12$ | 0.0091 | 0.05 | Reject H0 |
| $\gamma14$ | 0.7701 | 0.05 | Accept H0 |
| $\gamma15$ | 0.0129 | 0.05 | Reject H0 |
| $\gamma16$ | 0.6840 | 0.05 | Accept H0 |

Source: Own processing, Gretl.

In the first equation, both variables, namely yield of paddy in Vietnam and fertilizer use in rice production, are statistically significant toward the dependent variable, rice production in Vietnam. In the second equation, two variables out of the four, namely rice production in Vietnam and rice price in Thailand, are statistically significant toward the dependent variable, export amount of Vietnam's rice. The variables of rice price in Vietnam and rice consumption around the world are statistically insignificant.

The econometric verification is the last step. It is tested whether the model meets the assumptions of the classical linear regression model, which is essential before the model can be applied in practice. Several tests are conducted in order to detect autocorrelation, heteroscedasticity and normality of residual distribution. The verification for the first and second equation are shown in Tables 8 and 9 respectively.

**Table 8.** Econometric verification for the first equation.

| Test | *p*-Value | Significance Level | Hypothesis |
|---|---|---|---|
| Breusch–Godfrey Test | 0.0642 | 0.05 | Accept H0 |
| White Test | 0.237443 | 0.05 | Accept H0 |
| Frequency Distribution | 0.19378 | 0.05 | Accept H0 |

Source: Own processing, Gretl.

**Table 9.** Econometric verification for the second equation.

| Test | *p*-Value | Significance Level | Hypothesis |
|---|---|---|---|
| Breusch–Godfrey Test | 0.815 | 0.05 | Accept H0 |
| White Test | 0.183761 | 0.05 | Accept H0 |
| Frequency Distribution | 0.97324 | 0.05 | Accept H0 |

Source: Own processing, Gretl.

Autocorrelation occurs when the errors are correlated or dependent [10]. The Breusch–Godfrey test has been used to determine the occurrence of autocorrelation. The null hypothesis shows there is no autocorrelation. The *p*-value in both equations is higher than the significance level 0.05, meaning that the null hypothesis is accepted. This signals that the model does not have an autocorrelation problem. The existence of heteroscedasticity is a major concern in the econometric verification. It indicates that the modeling errors are no longer independently distributed. The White test has been used to determine the occurrence of heteroscedasticity. The null hypothesis says there is no heteroscedasticity. The *p*-value in both equations is higher than the significance level. As such, the null hypothesis is accepted, and homoscedasticity is assumed in both equations. The normality of residual distribution is key for all statistical modeling; otherwise, the results are not reliable. The null hypothesis shows that the residuals are distributed normally. Again, the *p*-value in both equations is higher than the significance level. The null hypothesis is accepted, thus indicating that the model has a normal distribution of residuals. The econometric verification confirms that the model meets the conditions necessary for application.

## 6. The Model's Application

Since the obtained estimates fulfill the expectations of the theory on which it is based and all the verification requirements are met, the model is suitable for practical application. An important area of the model's application is forecasting. The purpose is to estimate a value of the dependent variables outside the observation period and make a forecast. In particular, the amount of rice production and export in Vietnam between the years 2018–2023 is forecast. The resulting forecasts are listed in the Tables 10 and 11 shown below.

**Table 10.** Forecast for the production in 2018–2023.

| Vietnam's Rice Production (Thousand Tons) | | | | | |
|---|---|---|---|---|---|
| 2018 | 2019 | 2020 | 2021 | 2022 | 2023 |
| 45,827 | 46,143 | 47,381 | 47,741 | 48,847 | 49,461 |

**Table 11.** Forecast for the export in 2018–2023.

| Vietnam's Rice Export (Thousand Tons) | | | | | |
|---|---|---|---|---|---|
| 2018 | 2019 | 2020 | 2021 | 2022 | 2023 |
| 6022 | 5956 | 6254 | 6530 | 6972 | 6824 |

Visually, it is obvious that the estimated numbers are in line with the trend established in previous years. There is a steady growth in the production. As shown in Figure 3, the forecast predicts a growth

of 8% over the period, and it peaks with the produced quantity of 49,461 thousand tons in 2023, which is the biggest production quantity ever. Meanwhile, Vietnam is expected to face difficulties in exporting rice. The forecast predicts a stagnation in the exported quantity.

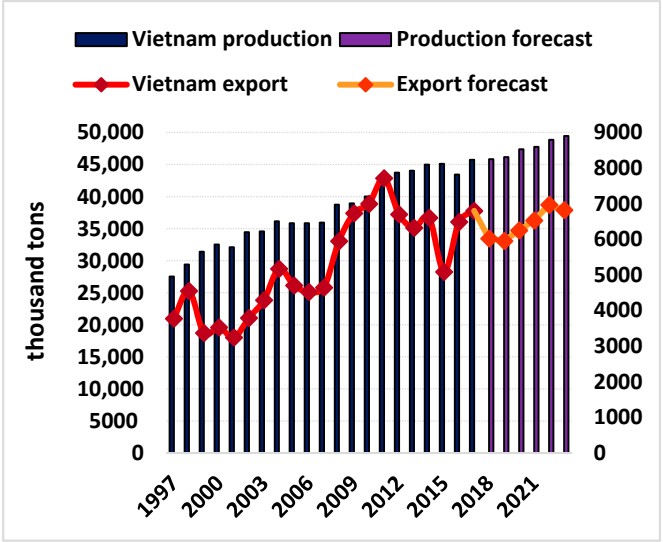

**Figure 3.** Forecast for the production and export in 2018–2023.

Vietnam's rice exports remain uncertain, especially because importing countries introduce policy changes on technical standards. In the past year, many rice-importing countries have undergone major policy changes, thus implementing rice tariffs and different ways in which sources participate in tenders to supply rice with competitive prices and a higher quality. Many countries have even made efforts to improve their domestic production capacity towards food autonomy [17–20]. Besides these difficulties, Vietnam also faces many new competitors in the market [18]. For example, China is Vietnam's biggest customer, but at the same time, it is expected that China will soon become a rice exporter and may even target Vietnam's markets, such as Africa, where cheap and low-quality rice dominates. To overcome these challenges, Vietnam needs to diversify its rice export markets and improve the quality of its rice so that it meets higher export standards.

## 7. Conclusions

Over the last few decades, rice has been Vietnam's critical crop for national food security. The government has played an active role in supporting rice production and increases in volume, first for the domestic market and later for exporting. The improvements in rice productivity have been the driving force behind the strong growth in its production and significant reductions in poverty. In two decades, from a country with food shortages, Vietnam has become one of the biggest rice exporters in the world. By all means, rice is no longer only an important staple food but also a strategic agricultural commodity in terms of international trade. In addition, activities relevant to rice production have been creating employment for millions of Vietnamese people.

Nevertheless, Vietnam's rice industry is facing many challenges. The limitations mainly lay in the old cultivation methods and low-quality rice varieties, which also negatively affect Vietnam's share in the international trade and the exported quantities. Finally, changes in import policies issued by Vietnam's major rice importers on the rice sector play a significant role.

**Author Contributions:** Conceptualization, K.M.; Data curation, K.M.; Formal analysis, M.M.; funding acquisition, M.M.; Investigation, K.M.; Methodology, L.S.; Resources, L.S.; Supervision, K.M. and J.S.; Validation, M.M.; Writing, M.M.; review & editing, J.S. and N.P.A. All authors have read and agreed to the published version of the manuscript.

**Funding:** This research received no external funding.

**Conflicts of Interest:** The authors declare no conflict of interest.

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
