# Peer review of "Rice as a Determinant of Vietnamese Economic Sustainability"

_sustainability, doi:10.3390/su12125123_

Round 1

Reviewer 1 Report

I have two comments for Authors.

1st. Please, explain why do you not include arable land as a variable for your model?

2nd. Labor, is a direct factor that has relation with production and farmers costs. The econometric model do not take this variable neither.

Author Response

Dear Respected Editors and reviewers:

We hereby confirm that the suggestions and comments addressed from the reviewers have been implemented. Furthermore, the authors have edited the manuscript for possible language changes, especially focusing on spelling, grammar, punctuation, and polishing.

The following points have addressed:

  1. We have updated the introduction, explaining the aim of the paper, the literature gap, and the motivation to conduct the study.
  2. We have provided an explanation on why the arable land is not used as a variable in the model.
  3. We have provided an explanation on why labor is not used as a variable in the model.
  4. We have added more resources.
  5. We have updated the in-text as well as reference list in accordance to the Instructions for Authors.
  6. We have added the text for the funding section.
  7. We have added the text for the conflict of interest section.
  8. The abstract of the paper was updated, reflecting changes in the introduction section.
  9. The manuscript was edited in terms of the English language.

Reviewer 2 Report

The manuscript presented for review is very interesting.

In my opinion, the paper is important for the world scientific society. The topic of the article is relevant and interesting.

The article needs strengthening in terms of Sustainability because after reading, I wondered why it was sent to the Sustainability Journal.

Please note and address the following comments:

Introduction: In my opinion, the introduction section should be changed, and introduce to economic sustainability concept in Vietnam, as well as answer why it is important for the Vietnamese agricultural sector.

Conclusion: Both in Introduction and in Conclusion, there is a lack of strengthening the justification of why this topic is so important. In my opinion, the introduction does not demonstrate a gap in the literature on why this research should be conducted. There is a lack of answers to the hypotheses in the conclusions.

Abstract: After making corrections, authors have to correct the abstract, which should correspond to added new information

I believe it addresses an important area of research in an international context.

Reviewer

Author Response

(The authors gave the same response as above.)

Round 2

Reviewer 1 Report

Some comments were included in this version

Reviewer 2 Report

Dear Authors

The authors have changed many parts of the planned paper according to my suggestions.

I would like to thank the authors for considering my comments and applaud them for the major revisions to improve their manuscript.

The manuscript is interesting and valuable.

Reviewer